# Ultra-Wideband Circular Polarized Implantable Patch Antenna for Implantable Blood Glucose Detection System Applications

**DOI:** 10.3390/s24165292

**Published:** 2024-08-15

**Authors:** Zhiwei Song, Mengke Li

**Affiliations:** The School of Electrical Engineering, Xinjiang University, Huarui Street 777#, Shuimogou District, Urumqi 830047, China; 107552304503@stu.xju.edu.cn

**Keywords:** antenna, circular polarization, continuous blood glucose monitoring systems, differential feed, miniaturization, ultra-wideband

## Abstract

To address the current demands for antenna miniaturization, ultra-bandwidth, and circular polarization in advanced medical devices, a novel ISM band implantable antenna for blood glucose monitoring has been developed. This antenna achieves miniaturization by incorporating slots in the radiation patch and adding symmetric short-circuit probes, resulting in a compact size of only 0.054λ_0_ × 0.054λ_0_ × 0.005λ_0_ (λ_0_ is the wavelength in free space in respect of the lowest working frequency). By combining two resonance points and utilizing a differential feed structure, the antenna achieves ultra-broadband and circular polarization. Simulations indicate a |S_11_| bandwidth of 1.1 GHz (1.65–2.75 GHz) and an effective axial ratio (based on 3 dB axis ratio) bandwidth of 590 MHz (1.89–2.48 GHz), able to cover both the ISM frequency band (2.45 GHz) and the mid-field frequency band (1.9 GHz). The antenna exhibits CP gains of −20.04 dBi at a frequency of 2.45 GHz, while it shows gains of −24.64 dBi at 1.9 GHz. Furthermore, a superstrate layer on the antenna’s radiating surface enhances its biocompatibility and minimizes its impact on the human body. Simulation and experimental results indicate that the antenna can establish a stable wireless communication link for implantable continuous blood glucose monitoring systems.

## 1. Introduction

With ageing and sub-health numbers rising, the demand for medical and healthcare services has also risen. As the traditional medical model makes it challenging to meet the growing demand for healthcare, wireless biomedical technology has emerged, providing a new direction for implantable electronic devices. These devices can break through the limitations of traditional medical treatment and provide more effective health protection [1].

Currently, complex implantable medical system devices have the advantage of uninterrupted monitoring of health data, predicting diseases and repairing human body functions compared with traditional medical devices. For example, implantable blood glucose monitoring instruments can detect real-time changes in blood glucose levels; cardiac pacemakers are used to treat cardiac dysfunctions such as arrhythmia [2]; and capsule endoscopy is used to take pictures by swallowing it into the digestive tract and transmitting the images outside of the body to determine the site and cause of a lesion [3,4,5,6,7,8]. However, the human body is a complex and diverse environment, which places high demands on the miniaturization and signal stability of implantable antennas. Large antennas can cause discomfort, so the design of miniaturized, wide bandwidth, and stable signal implantable antennas has become a hotspot in current research [9,10].

In [11], loading stacked parasitic loops and positioning metal plates on the reflector plate were used to expand the current path, thus designing a small antenna with a widened impedance bandwidth and a volume of 85 mm^3^. In [12], by incorporating metamaterial units, the researchers successfully miniaturized the antenna to a volume of 7.04 mm^3^. Therefore, these studies have effectively reduced the antenna size. However, since it is not a circular polarized (CP) antenna, further investigation into the stability of antenna signals is necessary. In [13], the authors designed a dual inductance CP microstrip patch antenna with a double coupled line, and achieved an axial ratio (AR) bandwidth of 0.33%. In [14], a novel single-fed CP microstrip antenna was designed, which produces an axial ratio bandwidth (ARBW) of 1.3% at 1.9 GHz. In [15], a CP filter patch antenna was proposed using a dispersion delay line (DDL), achieving a 3.8% 3 dB ARBW at 2.4 GHz. The aforementioned studies on circular polarization have led to significant improvements in signal reception, but the antennas’ axial ratio bandwidth remains suboptimal. In [16], the antenna has a size of 257 mm^3^ and achieves CP through orthogonal feeding elements. It operates at 2.4 GHz and 5.8 GHz and demonstrates UWB characteristics. In [17], the authors used four L-shaped slots and two orthogonal rectangular slots on the radiating surface to achieve a wide bandwidth, including S_11_ and AR from 2.3 to 2.8 GHz and 2.3 to 2.6 GHz, respectively. The antenna volume was 63.83 mm^3^. The aforementioned research on wide bandwidth has significantly expanded the antenna’s frequency range, yet there is still potential for further reductions in the antenna volume.

To take into account the miniaturization, circular polarization, and wide bandwidth of antennas for complex medical system devices, this paper proposes an UWB implantable CP patch antenna designed for blood glucose monitoring, with the antenna resonant frequency in the 2.45 GHz frequency band. By using a high dielectric constant material and the slotting method, we successfully miniaturized the antenna, and realized the UWB characteristics of the antenna by combining multiple resonance points through the slotting, as well as adopting a differential-feeding structure and a pair of symmetrical short-circuiting probes to realize circular polarized antenna performance whilst maintaining miniaturization and UWB.

## 2. Design Methodology

### 2.1. Antenna Configurations

The antenna’s structure was very simple. From its top to bottom, there is a superstrate layer, a radiation patch, a dielectric substrate, and a ground plane, as shown in Figure 1c. The detailed geometric variables and their sizes are listed in Table 1. The radial plane has an inside-out annular groove and three notched annular grooves. There is a combination groove in the ground plane, which consists of three horizontal rectangular grooves and one vertical rectangular groove, which significantly reduces the antenna size, increases the bandwidth, and enables frequency tuning. The dielectric substrate is made of Rogers 6010 with a profile of 0.254 mm. Since the fabrication process of the selected board is not within the scope of this paper, this is not elaborated in detail. The antenna is fed by two ports with an inner diameter of 0.25 mm and the feed position is optimized for best impedance matching and broadband performance. To prevent direct contact with human tissue, a superstrate layer is added. This layer is made of Rogers 6006 material, which has a relative permittivity (ε_r_) of 6.15. The antenna is optimized to superstrate the 2.4–2.48 GHz band and is compact with dimensions of 0.054λ_0_ × 0.054λ_0_ × 0.005λ_0_.

In the simulation, we used a single-layer human skin tissue to simulate the actual application environment of the antenna, and its size was 89.8 × 89.8 × 26.27 mm^3^ (ε_r_ = 38, σ = 1.44 S/m, tanδ = 0.283). Figure 2 illustrates the complex medical system integration device. Direct contact between the antenna and the human tissue impacts antenna performance. During the simulation, the battery was set as a perfect electrical conductor (PEC), while all other circuit components were set as Rogers RT/duroid 3010, and the dimensions of the different components in Figure 2 are the sensor set (π × 3 mm × 2.6 mm), the PCB unit (12 mm × 10 mm × 0.5 mm), and the battery (5 mm × 10 mm × 2.7 mm). Figure 3a compares the antenna’s reflection coefficient (|S_11_|) versus frequency curves before and after adding a superstrate layer. The simulation results clearly demonstrate that the antenna’s performance significantly improves with the addition of the dielectric superstrate layer, enhancing the bandwidth and fully meeting the requirements of the ISM band. The reason is that the introduction of the superstrate layer not only reduces the fringing effect of the antenna, but also enhances the matching performance, restricts the unintended radiations, and strengthens the adaptability to the environmental change and thus the |S_11_| parameters achieve the desired effect. Since, when loading the antenna with a superstrate layer, a small gap between the antenna superstrate layer and the radiating patch is inevitable because of manual operation, the variation in the antenna’s |S_11_| with the gap between the superstrate layer and the radiating patch is further shown in Figure 3b, which shows that the antenna’s |S_11_| also fluctuates slightly with the increase in the gap but the overall situation still superstrates the bandwidth of the desired frequency band. When placing the antenna inside the integrated device, the effect of the components inside the device on the antenna performance needs to be considered, and the benefits or drawbacks of this effect are uncertain and need to be experimentally demonstrated. As shown in Figure 3c, there is a small increase in bandwidth after placing the antenna in the integrated device, indicating that the addition of the integrated device improves the performance of the antenna. In the process of antenna design and optimization, the above factors are fully considered, which plays an essential role in improving the antenna performance and its operating characteristics.

### 2.2. Antenna Working Principles

Figure 4a illustrates the process of UWB. Figure 4b shows the evolution of the dual-band CP antenna. Figure 5 shows a comparison of the results of each step.

Step 1: A feed port is introduced on top of the radiation patch. As shown in Figure 4a, by introducing two short pins connecting the radiating element of the antenna and the ground plane, the current flow path is changed, and the current flows directly to the ground through the short pins, so that a part of the antenna’s electrical length is ‘collapsed’. This reduces the effective length of the antenna, and realizes the miniaturization of the antenna.

Step 2: A hollow circular ring and a hollow C-shaped open ring have been slotted in the radiation patch. Additionally, a new feeder is introduced. The new structure excites an additional resonance point at 1.5 GHz compared to step 1, as shown in Figure 5a. The axial ratio (AR) of the new structure is also better than that of step 1, as shown in Figure 5b (step 2), indicating that the introduction of differential feeding can be used to obtain the stable 90° phase difference required for circularly polarized radiation by feeding differential signals of equal amplitude and opposite phase through the two ports, which are spatially orthogonal to each other, resulting in circular polarization. The introduction of the slot structure changes the distribution and path of the current on the surface of the antenna, leading to a change in the path of the current at different frequencies, which generates a new TM mode, allowing the antenna to form a resonance at more than one frequency, which is more conducive to the dual-frequency design.

Step 3: Two C-shaped slots and a notched annular slot are introduced on the radiation patch based on step 2, as shown in Figure 4c. This radiating surface structure changes the current path as shown in step 3 in Figure 4a. Thus, the two bands are merged into a single operating band, realizing the UWB of the antenna. Generated 2.10 GHz to 2.94 GHz |S_11_|, successfully superstrating the ISM band. This is achieved by the bending of the radiating unit’s surface, which extends the effective current path. This adjustment lowers the resonance frequency of the implanted antenna. Additionally, it merges the two resonance points to achieve wide bandwidth characteristics. However, the ARBW is insufficient to superstrate the ISM band.

Step 4: The design of step 3 is further improved using a two-port differential feed to provide a stable 90-degree phase difference, which continues to widen the bandwidth. Additionally, a combination groove is introduced, reducing the antenna’s resonant frequency, shifting the two resonance points of the antenna towards the lower frequencies, thus superstrating the desired band and increasing the ARBW. This allows the antenna to achieve UWB CP radiation characteristics with an effective 3 dB ARBW of 590 MHz (1.89 GHz–2.48 GHz) as shown by the red line in step 4 of Figure 5b. The ARBW is significantly improved. CP is achieved in the 1.9 GHz band. Additionally, CP is also achieved in the 2.45 GHz band.

The mechanism of production CP: Figure 6 illustrates the simulated AR of the optimized antenna. The dual-band CP mechanism can be investigated by studying the current distribution. Figure 7 illustrates the current vector distribution of the antenna in four phases at two operating frequencies, as shown in Figure 5a (red line). The arrows in the Figure 7 indicate the direction of the current vector sums for different phases. The current rotates counterclockwise in these phases, indicating right-handed circular polarization (RHCP), as shown in Figure 8.

In order to analyze the immunity of the antenna impedance detuning to changes in the electromagnetic properties of the surrounding tissue, the antenna matching levels were analyzed in a rectangular model of 89.8 mm × 89.8 mm × 26.27 mm with $\varepsilon_{r}\in [25,60]$ and σ∈[0,3] at 1.9 GHz and 2.45 GHz [18]. The range includes biological tissues implanted at 1.9 GHz and 2.45 GHz. Figure 9 shows the detuned immunity range of the antenna. It is clear from the figure that the antenna’s |S_11_|, gain, and axial ratio (based on 3 dB) show good antenna performance in the implanted tissues. Meanwhile, the antenna can still operate normally when small variations in dielectric constant and conductivity occur, confirming the stability of the antenna operation.

### 2.3. Radiation Patterns, Gain and SAR

Figure 8 shows the RHCP and left-handed circular polarization (LHCP) gain plots of the designed CP antenna at *xoy*-plane and *xoz*-plane. In the single-layer human tissue model, the antenna produced a gain value of −20.04 dBi in the ISM band and −24.64 dBi in the mid-field band. The radiation values are measured in the direction towards the outside of the human body. Owing to implantation in human tissue with loss and dispersive properties, negative gain values are achieved. The RHCP radiation, shown in Figure 8, is consistent with the results in Figure 7.

Since the antenna needs to be implanted into human tissue, its radiation will interact with the tissue, so the specific absorption rate (SAR) must be taken into account to ensure human safety. The calculation formula is as follows [19]:(1)$$SAR=\frac{\sigma E^{2}}{2\rho }$$
where *E* is the electric field strength and *ρ* is the tissue density.

The simulated 1g-averaged SAR values are 421 W/kg (1W) at 1.9 GHz. At 2.45 GHz, the SAR value is 463 W/kg (1W). To meet the 1999 IEEE standard for 1g-averaged SAR, the input power should not exceed 3.7 mW at 1.9 GHz and 3.4 mW at 2.45 GHz [20,21]. These values indicate that there is no harm to the patient’s body within the permitted input power range. The simulated SARs are depicted in Figure 10.

## 3. Link Budgets

To ensure the reliable and efficient transmission of physiological data between the device and external devices, the transmission distance is calculated according to Equations (2)–(4) [22]. To ensure safety and stability, the input power must not exceed −46 dBw, taking into account the limits of the European Radio Communications Commission (ERC). The link budget equation is as follows:(2)$$LM(dB)=\frac{C_{1}}{N_{0}}-\frac{C_{2}}{N_{0}}$$
(3)$$\frac{C_{1}}{N_{0}}=P_{t}+G_{t}+G_{r}-L_{f}-N_{0}$$
(4)$$\frac{C_{2}}{N_{0}}=\frac{E_{b}}{N_{0}}+10\log(B_{r})-G_{c}+G_{d}$$

Figure 11 illustrates the variation in the transmission distance for both frequencies at 100 kbps, the link margins remain robust at 37.74 dB and 35.35 dB for transmissions up to 20 m. However, increasing the bitrate to 1 Mbps reduces the link margins to 27.74 dB and 25.35 dB at the same transmission distance [23,24,25,26,27,28,29,30,31]. The parameters used in the link budget are tabulated in Table 2.

## 4. Antenna Tests and Discussion of Results

From the data listed in Table 1, the antenna was processed and the performance of the antenna was verified experimentally. The antenna is tested with two coaxial cables, connected to two excitation source ports, where the ports are set up with a current phase difference of 90° as a means of differentially feeding the antenna. Figure 12 illustrates the antenna prototype and the measurement setup, with |S_11_| testing performed using a vector network analyzer in minced pork. The size of the container for storing the pork puree is 25 cm × 18 cm × 10 cm, and the implantation depth of the complex implantable blood glucose monitoring system is 4 mm. Figure 13 compares the |S_11_| results of the antenna in the simulated and measured cases. The red curve represents measured results in minced pork, while the blue curve shows simulation results from software. The impedance bandwidth tested in minced pork environment spans approximately 54.8% (1.63–2.86 GHz). A slight deviation of |S_11_| from the HFSS simulation is observed, likely attributed to machining tolerances and measurement errors. Despite minor differences, both results encompass the desired ISM band.

Figure 14 shows the AR values in different environments. In the simulated scenario, the 3 dB ARBW spans 0.59 GHz (1.89 GHz–2.48 GHz), while in the minced pork, the measured ARBW is 0.58 GHz (1.9 GHz–2.48 GHz). The experimental results indicate that the discrepancy between simulation and measurement is negligible. Typically, such errors stem from differences between the test and simulation environments, as well as minor variations in antenna manufacturing and operational procedures.

Figure 15 and Figure 16 depict the gain results of the designed antenna in both experimental environments. For clarity, Figure 17 presents the three-dimensional radiation patterns. It should be noted that in both the simulation and the experiment, we conducted research on implantable blood glucose monitoring systems. For the sake of observation, in the simulation process of Figure 17, we have hidden the remaining parts in HFSS and only displayed the antenna part. A comparison is conducted between the antenna’s gain patterns simulated in a standard environment and those measured in the minced pork medium. It is observed that the antenna’s main radiation direction is outward. In the single-layer skin model, peak gains at 2.45 GHz and 1.90 GHz are measured at −20.04 dBi and −24.64 dBi, respectively.

In the minced pork environment, these peak gains shift slightly to −21.33 dBi and −26.07 dBi at the same frequencies. These results indicate a minor decrease in gain across different environments, likely attributed to variations in dielectric constants and other parameters between minced pork and human skin, as well as differences in implantation depth. Nevertheless, the antenna maintains good gain values, meeting the expected performance criteria.

In order to better represent the performance of the designed antenna, at the end of this article, the designed antenna is compared with the antennas in the literature. Table 3 makes a comparison between this paper and the references in terms of tissue properties (dielectric constant and conductivity), shape, and tissue size. Table 4 makes a comparison among this paper and the references in terms of radiation properties and physical characteristics (center frequency, FBW, ARBW, gain, polarization characteristics, and size). From Table 4, the antenna presented in this paper is significantly optimized in terms of size and achieves UWB effective ARBW, whereas the ARBW in Table 4 is calculated based on 3 dB. However, its gain performance is not the highest among the compared antennas.

## 5. Conclusions

In this paper, a UWB CP implantable patch antenna is designed for blood glucose monitoring in complex medical devices. It is shown that the bandwidth and CP characteristics of the antenna are achieved by differential feeding, introducing short-circuiting probes, and etching two C-shaped slots, a ring slot and a notched ring slot, while combining the multiple resonance points on the backside by opening a combination groove. A satisfactory 3 dB axial bandwidth of 24.1% and an impedance bandwidth of 44.9% are achieved. In addition, the antenna in a real organizational environment confirms that the antenna has better robustness. The link margin and SAR distribution are also evaluated. Finally, the antenna prototype is tested in minced pork to measure its radiation characteristics, and the tested |S_11_| and CP follow the expected results of the simulation.

## Figures and Tables

**Figure 1 sensors-24-05292-f001:**
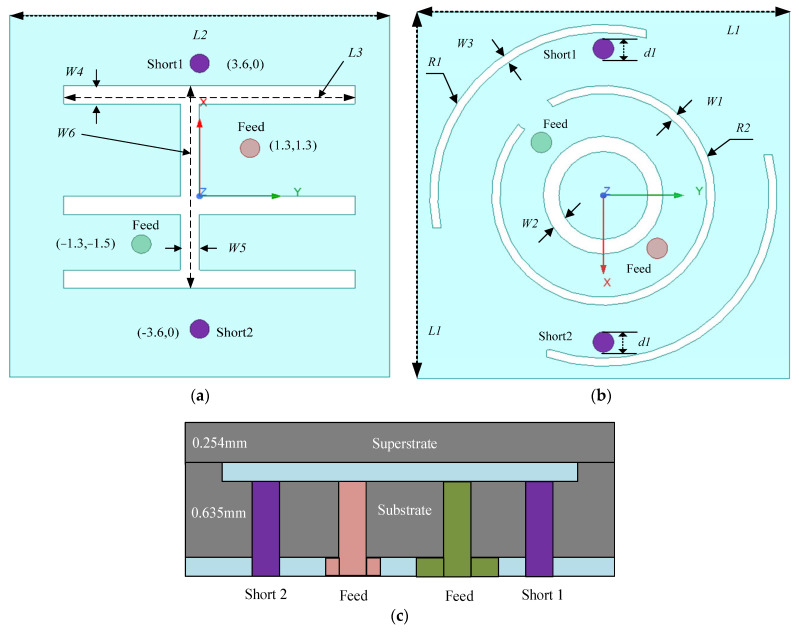
Geometry of the designed antenna. (**a**) Bottom view; (**b**) top view; (**c**) side view (the gap between the superstrate and antenna layer is 0 mm).

**Figure 2 sensors-24-05292-f002:**
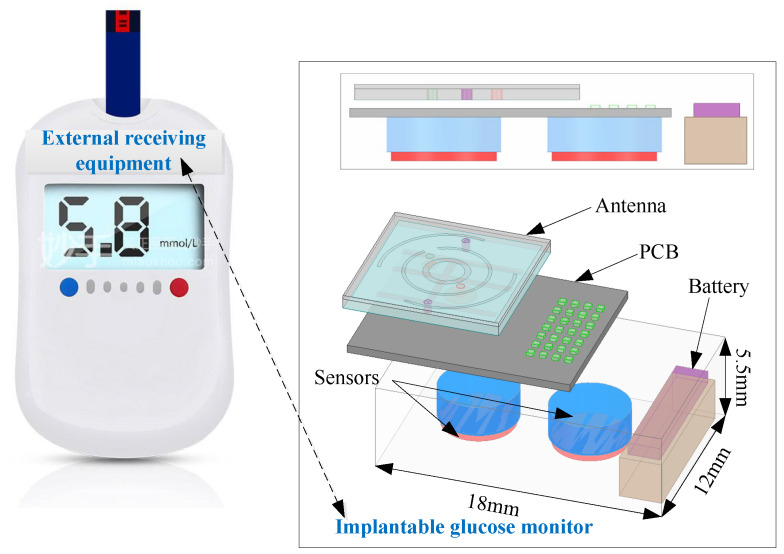
Implantable device and external receiving equipment.

**Figure 3 sensors-24-05292-f003:**
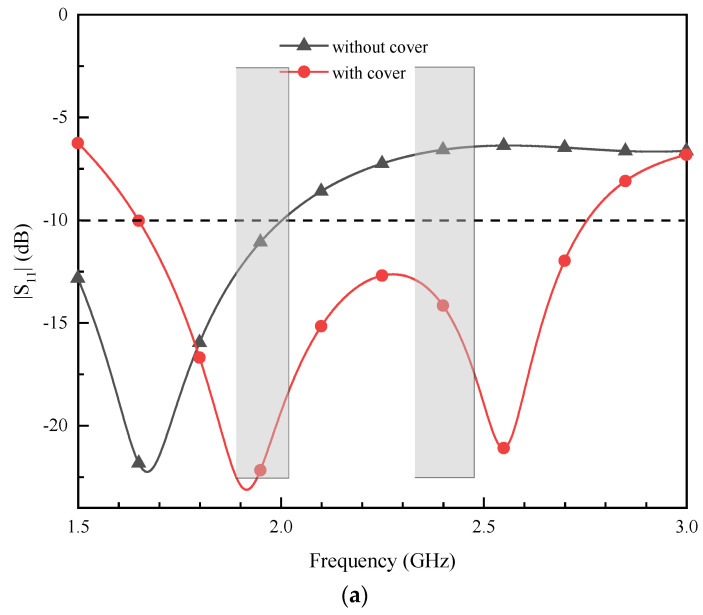
(**a**) Comparison of the simulated |S_11_| when the antenna with and without the superstrate; (**b**) variation in |S_11_| with superstrate layer to radiating patch gap (gap = 0 mm is the ideal case); (**c**) comparison chart with and without integrated device |S_11_|.

**Figure 4 sensors-24-05292-f004:**
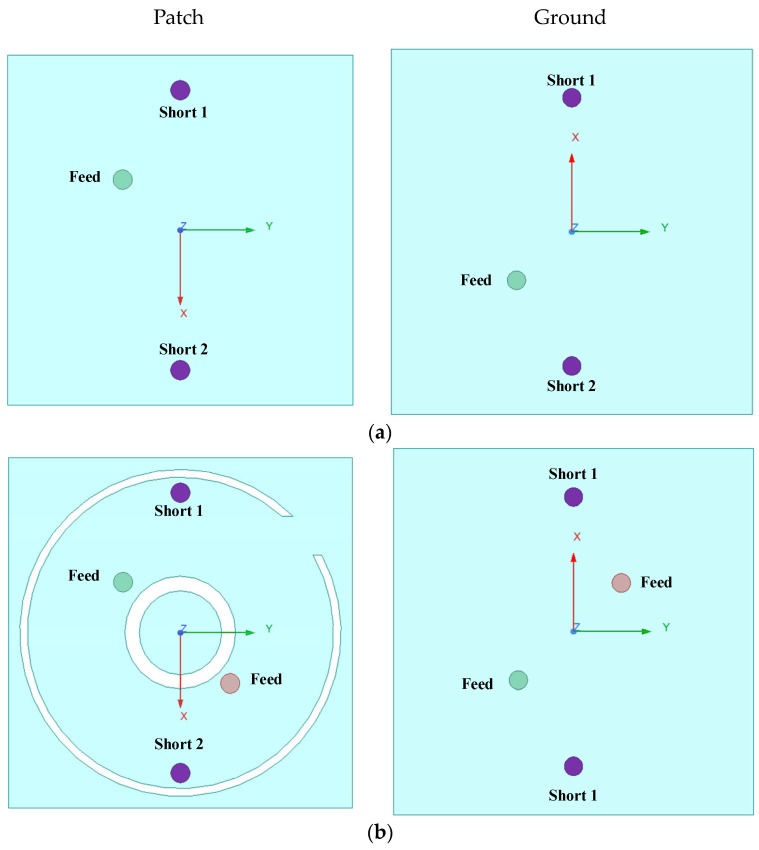
Antenna design procedure. (**a**) Step 1; (**b**) step 2; (**c**) step 3; (**d**) step 4.

**Figure 5 sensors-24-05292-f005:**
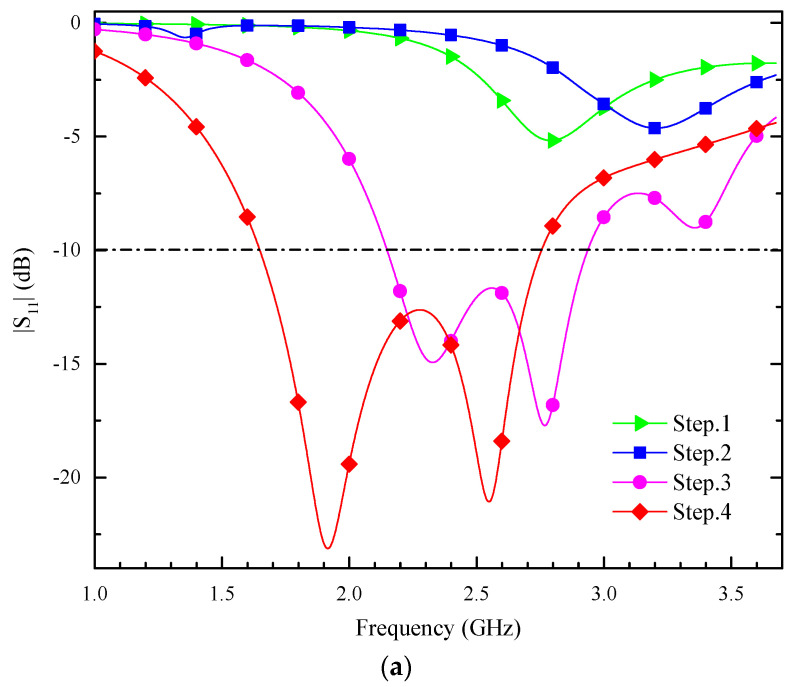
The simulated (**a**) |S_11_| and (**b**) AR in antenna evolution process.

**Figure 6 sensors-24-05292-f006:**
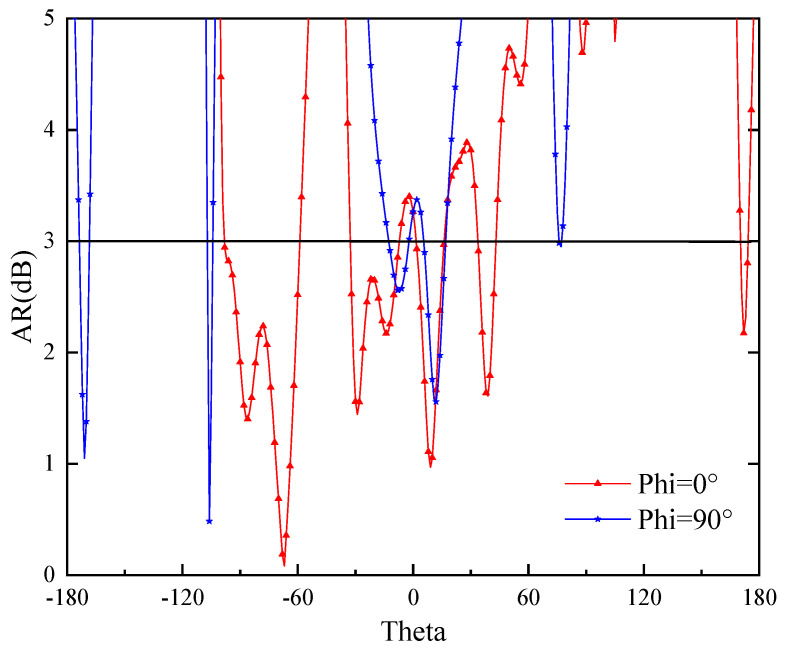
The simulated AR of the optimized antenna. Phi = 0° and Phi = 90°.

**Figure 7 sensors-24-05292-f007:**
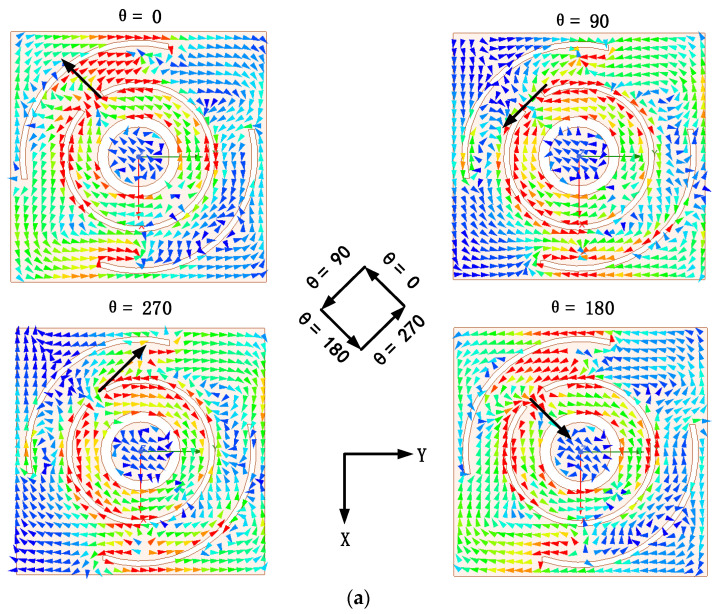
Current vector distribution on antenna radiation surface. (**a**) 1.9 GHz; (**b**) 2.45 GHz.

**Figure 8 sensors-24-05292-f008:**
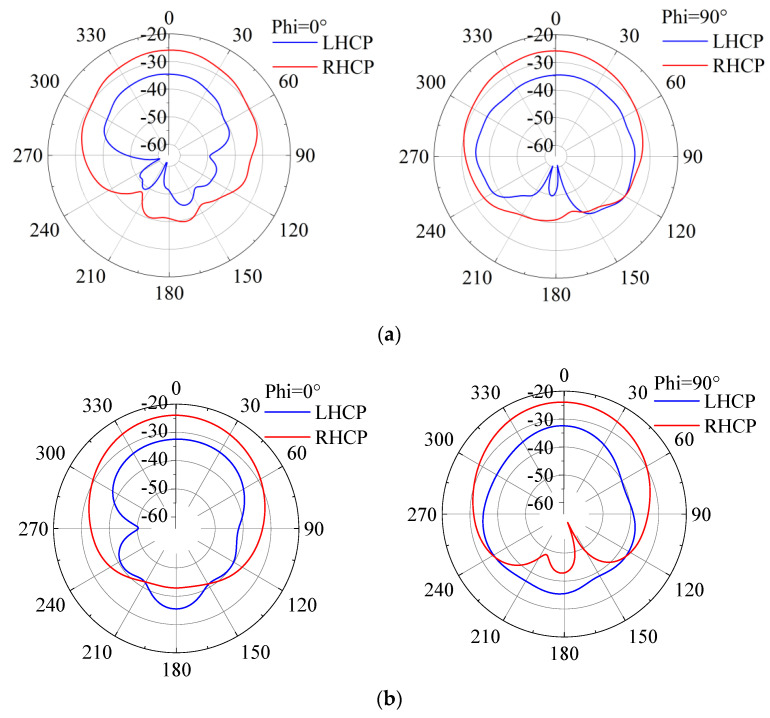
The simulated CP gain at *xoy*-plane and *xoz*-plane. (**a**) 1.9 GHz; (**b**) 2.45 GHz.

**Figure 9 sensors-24-05292-f009:**
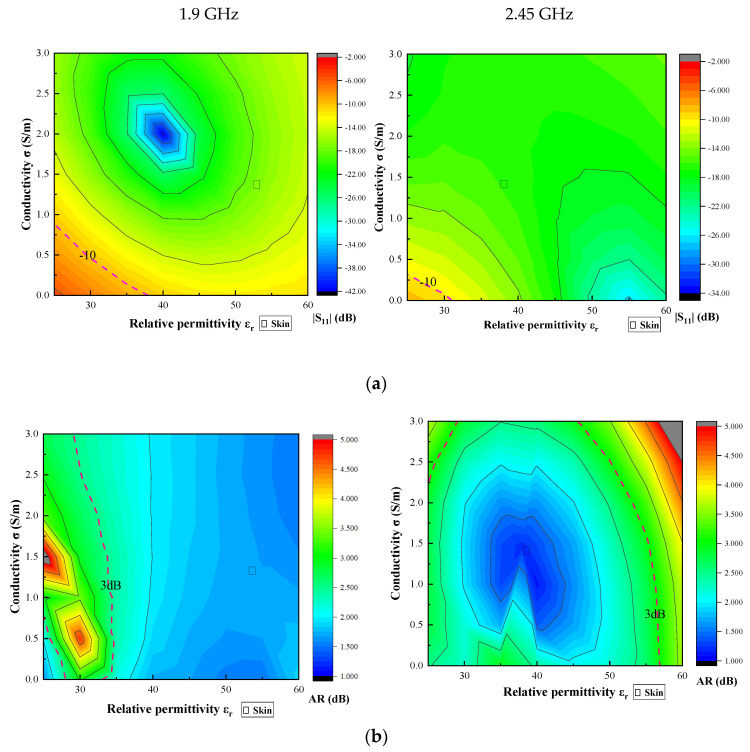
Antenna detuning immunity. (**a**) |S_11_|; (**b**) AR; (**c**) Gain.

**Figure 10 sensors-24-05292-f010:**
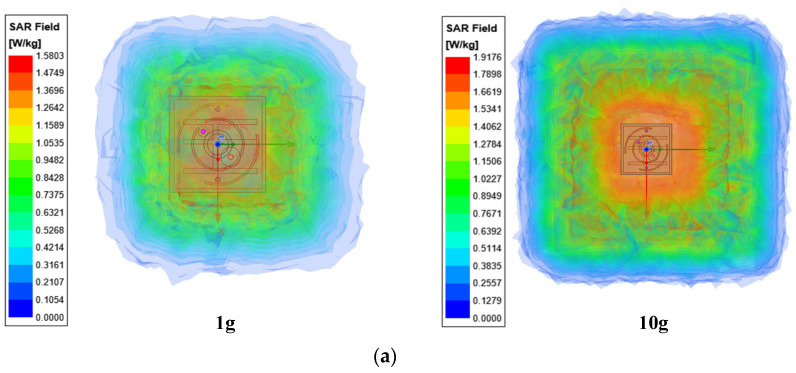
Analogue SAR (W/kg) in single-skinned human tissue and integrated packages. (**a**) 1.9 GHz (human tissue); (**b**) 2.45 GHz (human tissue); (**c**) 1.9 GHz (integrated package); (**d**) 2.45 GHz (integrated package).

**Figure 11 sensors-24-05292-f011:**
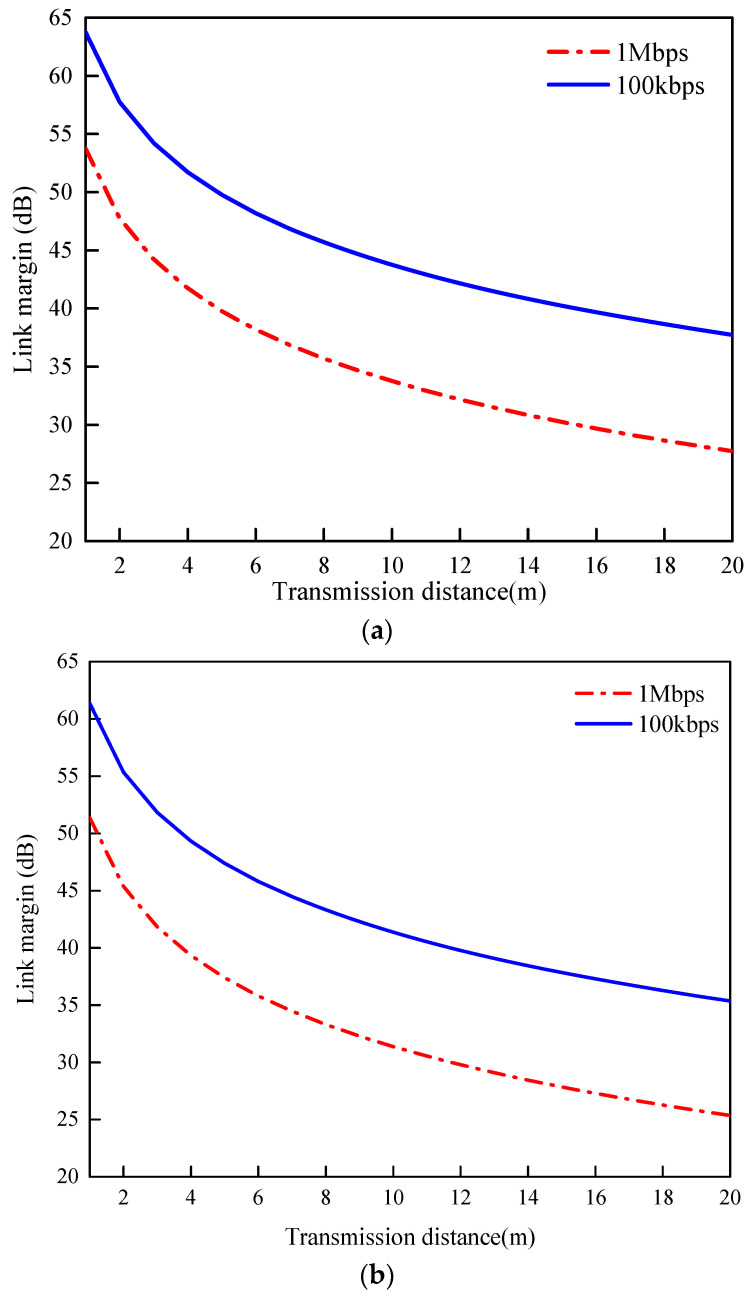
Link margin versus distance for different frequencies. (**a**) 1.9 GHz; (**b**) 2.5 GHz.

**Figure 12 sensors-24-05292-f012:**
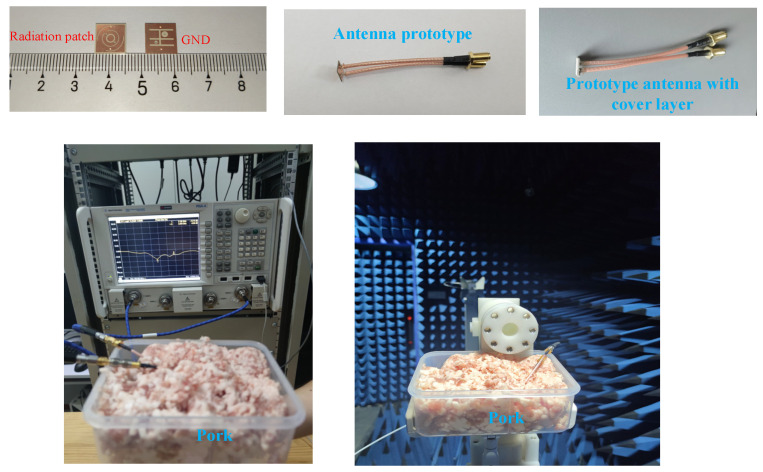
Antenna prototype and test scenario.

**Figure 13 sensors-24-05292-f013:**
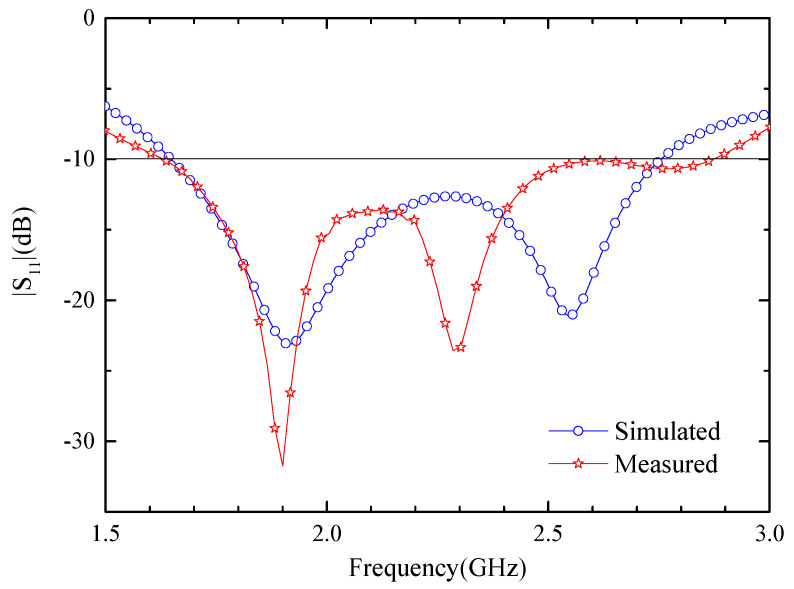
Antenna| S_11_| results in simulated and measured cases.

**Figure 14 sensors-24-05292-f014:**
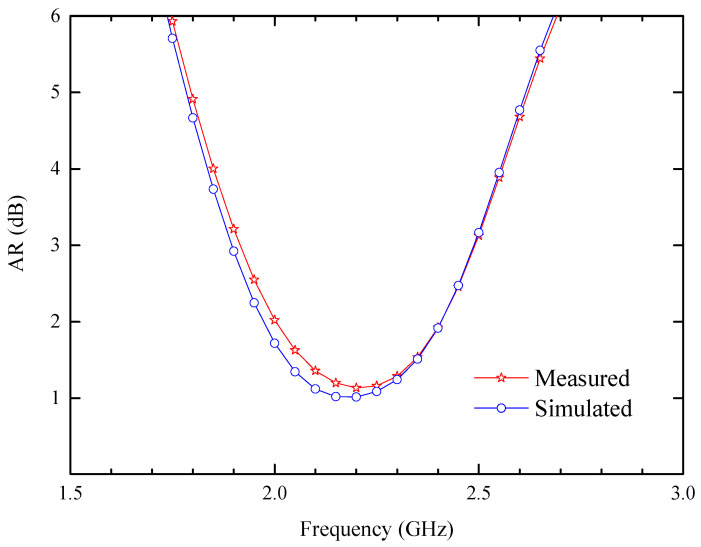
Comparison of the axial ratio between the simulated results and measured results.

**Figure 15 sensors-24-05292-f015:**
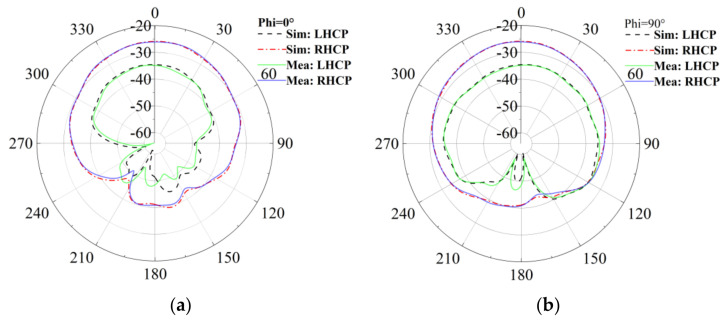
The circularly polarized gain measurements compared to simulated results of the antenna at 1.9 GHz. (**a**) Phi = 0°; (**b**) Phi = 90°.

**Figure 16 sensors-24-05292-f016:**
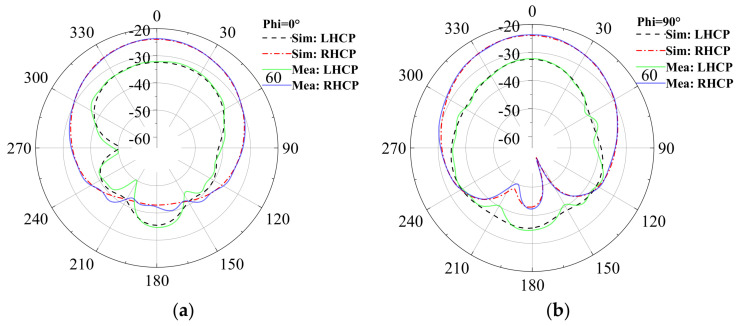
The circularly polarized gain measurements compared to simulated results of the antenna at 2.45 GHz. (**a**) Phi = 0°; (**b**) Phi = 90°.

**Figure 17 sensors-24-05292-f017:**
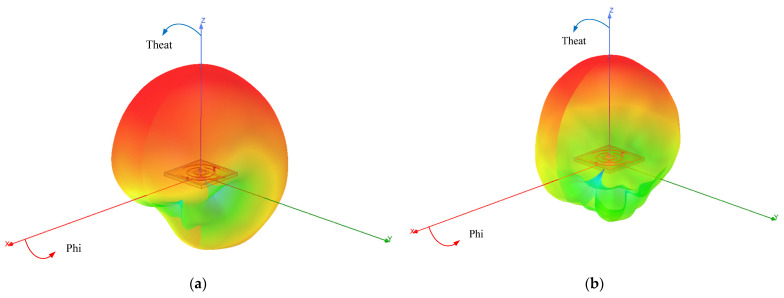
The simulated 3D radiation patterns of the designed antenna. (**a**) 1.9 GHz; (**b**) 2.45 GHz.

**Table 1 sensors-24-05292-t001:** Key parameters of the designed antenna and their sizes, and all sizes are in mm.

Parameter	Value	Parameter	Value
*L1*	9.0	*W4*	0.5
*L2*	9.8	*W5*	0.5
*L3*	7.5	*W6*	5.5
*W1*	0.2	*R1*	4.2
*W2*	0.379	*R2*	2.7
*W3*	0.2	*d1*	0.5

**Table 2 sensors-24-05292-t002:** Parameters of the link budget.

Transmitter		
Operating frequency	1.9 GHz	2.45 GHz
Tx power *P_t_* (dBm)	4.24	5.02
Tx antenna gain *G_t_* (dBi)	−18.8	−19.1
Receiver		
Rx antenna Gain *G_r_* (dBi)	2.15	2.15
Polarization	CP	CP
Ambient temperature *T*_0_ (K)	293	293
Receiver noise figure NF (dB)	3.5	3.5
Boltzmann constant *k*	1.38 × 10^−23^	1.38 × 10^−23^
Noise power density (dB/Hz)	−199.95	−199.95
Signal quality		
Bite rate *B_r_* (Mb/s)	1	1
Bite error rate	1.0 × 10^−5^	1.0 × 10^−5^
*E_b_*/*N*_0_ (ideal-BPSK) (dB)	9.6	9.6
Coding gain (dB)	0	0
Fixing deterioration *G_d_* (dB)	2.5	2.5

**Table 3 sensors-24-05292-t003:** Tissue properties, size, and shape.

Ref.	Tissue	Size (mm)	Frequency (GHz)	Permittivity (ε_r_)	Conductivity S/m (σ)	Shape
[5]	skin, fat	30 × 30 × 5.2	2.45	5.2749	0.26	cuboid
[19]	skin, fat, muscle	130 × 130 × 45.27	2.45 1.90	5.28 53.4	0.26 1.39	cuboid
[23]	skin, fat, muscle	100 × 100 × 100	2.45	52.7	1.74	cube
[24]	skin	100 × 100 × 100	2.45	38	1.46	cube
[25]	skin skin	120 × 120 × 40	2.45 0.915	38 11.3	1.46 0.11	cuboid
[26]	skin	100 × 100 × 100	2.45	42.92	1.56	cube
This work.	skin	89.8 × 89.8 × 26.27	1.90 2.45	53.4 38	1.39 1.46	cuboid

**Table 4 sensors-24-05292-t004:** Comparison of this paper with the literature.

Ref.	Size (λ_0_^3^)	Frequency (GHz)	FBW(%)	3-dB AR (%)	Gain (dBi)	CP	Tissue
[5]	0.167 × 0.291 × 0.03	2.45	6.7	no	−2.0	no	fat
[20]	0.08 × 0.08 × 0.01	2.45 1.9	18.7 7.5	17.26 1.37	−18.5 −21.8	yes yes	fat muscle
[24]	0.09 × 0.047 × 0.002	2.45	6.1	no	−22.2	no	muscle
[25]	0.08 × 0.08 × 0.01	2.45	30	28.7	−24.7	yes	skin
[26]	0.103 × 0.103 × 0.01	2.45 0.915	10.2 16.4	no no	−21.2 −30.3	no no	skin
[27]	0.13 × 0.07 × 0.03	2.45	14.3	no	−20.0	no	skin
This work.	0.054 × 0.054 × 0.01	1.90 2.45	57.9 44.9	31.4 24.1	−24.6 −20.0	yes yes	skin

## Data Availability

Data are contained within the article.

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
