# Peer review of "Ultra-Wideband Circular Polarized Implantable Patch Antenna for Implantable Blood Glucose Detection System Applications"

_sensors, 2024, doi:10.3390/s24165292_

Round 1
Reviewer 1 Report
Comments and Suggestions for Authors
In this manuscript, the authors propose an ultra-wideband circularly polarized implantable patch antenna by applying short pins, differential feed structure, and groove in the ground plane.
I agree with much of the manuscript's approach, including the design procedure, SAR and link budget calculation, and measurement method. However, I believe that in order to be published in the journal ‘Sensors’, some contents of the manuscript should be clarified and revised.
1. Some contents of the manuscript need to be revised for readers’ understanding.
1) Please indicate which part is the antenna in Figure 1(c).
2) Are the ‘cover layer’ and ‘encapsulation layer’ the same?
3) Is the ARBW in Table 3 based on 3 dB?
4) Does ‘reducing’ in line 3 of Step 4 means ‘downshift’?
5) Please check the paragraph format in Section 2.1.
2. Please provide justification (reference from previous studies or physical description) for the techniques used in steps 1-4 in Section 2.2.
1) Antenna miniaturization by short pins.
2) CP design by differential feeding.
3) Dual frequency by slotted structures.
4) Increased bandwidth by introducing grooves on the ground plane.
3. In Table 3, please compare the results of this manuscript’s results with recent papers (within 3 years).
4. In Figure 7, how did the authors obtain the directions of current? It seems to me that the direction of the current indicated by the authors is somewhat different from the direction of the simulated current.
5. According to Chu’s limit, as the size of the antenna increases, the Q factor decreases and BW increases, as shown in the equation below:
Q = 1/(ka) + 1/(k3a3)
BW = 1/Q = (f2-f1)/fc*.
According to my calculations, the theoretical bandwidth limits of the antennas in Table 3 are as follows (calculated based on the diagonal size of the antenna):
[3] = 152.7%
[5] = 172.0%
[12] = 162.9%
[13] = 367.4%
[14] = 184.7%
[15] = 563.9%
This work. = 8.9%.
However, the antenna proposed in this manuscript far exceeds the bandwidth of Chu’s limit.
Reviewer 2 Report
Comments and Suggestions for Authors
This paper shows an UWB antenna for medical applications. There are several questions as follows:
1. In Fig. 3, antenna with cover can exhibit better matching performance. Pls explain this phenomenon clearly. Hows the cover layer affect the antenna input impedance.
2. In Step1, the shorting pins are introduced. Pls comment hows much size reduction.
3. In Step 4, the defected ground structure is used. This definitely causes negative effect on the antenna radiation by significantly increasing the back radiation. Pls comment on this aspect.
4. In Fig. 6, the AR values are less than 3 dB. However, the difference between the RHCP and LHCP in the broadside direction is minor, just 5 dB. It is somehow unreasonable. Pls check it.
Comments on the Quality of English LanguageEnglish is good.
Reviewer 3 Report
Comments and Suggestions for Authors
My comments to the authors can be found in the attached document.

Moderate editing of English language required.
Round 2
Reviewer 2 Report
Comments and Suggestions for Authors
The authors have addressed all concerns properly.
Author Response
Your suggestion is very helpful in improving the quality of the paper. Thank you for your hard work and recognition of our work.
Reviewer 3 Report
Comments and Suggestions for Authors
There are still several questions and issues that need further clarification and improvement. Please find them in the attached document.

It can be improved
